# Development and cross-validation of a veterans mental health risk factor screen

Eve B. Carlson[1,2]*, Patrick A. Palmieri[3], Dawne Vogt[4,5], Kathryn Macia[1,2], Steven E. Lindley[2,6]

1 Dissemination and Training Division, National Center for Posttraumatic Stress Disorder, VA Palo Alto Health Care System, Menlo Park, California, United States of America, 2 Psychiatry and Behavioral Sciences, Stanford University School of Medicine, Stanford, California, United States of America, 3 Traumatic Stress Center, Department of Psychiatry, Summa Health System, Akron, Ohio, United States of America, 4 Women's Health Sciences Division, National Center for Posttraumatic Stress Disorder, VA Boston Healthcare System, Boston, Massachusetts, United States of America, 5 Department of Psychiatry, Boston University School of Medicine, Boston, Massachusetts, United States of America, 6 VA Palo Alto Health Care System, Palo Alto, California, United States of America

* ecarlson@stanford.edu

## Abstract

### Background

VA primary care patients are routinely screened for current symptoms of PTSD, depression, and alcohol disorders, but many who screen positive do not engage in care. In addition to stigma about mental disorders and a high value on autonomy, some veterans may not seek care because of uncertainty about whether they need treatment to recover. A screen for mental health risk could provide an alternative motivation for patients to engage in care.

### Method

Data from samples of veterans and traumatic injury survivors were analyzed to identify mental health risk factors that are characteristics of individuals or stressors or of post-trauma, post-deployment, or post-military service resources, experiences, or responses. Twelve risk factors were strongly related to PTSD (r > .50): current PTSD, depression, dissociation, negative thinking, and emotional lability symptoms, life stress, relationship stress, social constraints, and deployment experiences of a difficult environment, concerns about life and family, perceived threat, and moral injury. Items assessing each of these risk factors were selected and their validity to prospectively predict PTSD and/or depression 6 months later was assessed in a new sample of 232 VA primary care patients.

### Results

Twelve items assessing dissociation, emotional lability, life stress, and moral injury correctly classified 86% of those who later had elevated PTSD and/or depression symptoms (sensitivity) and 75% of those whose later symptoms were not elevated (specificity). Performance was also very good for 110 veterans who identified as members of ethnic/racial minorities.

**Data Availability Statement:** All data files are available from the Dryad database here: https://datadryad.org/stash/dataset/doi:10.5061%2Fdryad.tb2rbp03g.

**Funding:** E.B.C. and P.A.P. were funded with a Merit Grant by the Clinical Services Research and Development program of the Department of Veterans Affairs. https://www.research.va.gov/services/csrd/default.cfm The funders had no role in study design, data collection and analysis, decision to publish, or preparation of the manuscript.

**Competing interests:** The authors have declared that no competing interests exist.

## Conclusions

Mental health status was prospectively predicted in VA primary care patients with high accuracy using a screen that is brief, easy to administer, score, and interpret, and fits well into VA's integrated primary care. When care is readily accessible, appealing to veterans, and not perceived as stigmatizing, information about mental health risk may result in higher rates of engagement than information about current mental disorder status.

## Introduction

Military personnel who are deployed to a warzone experience high levels of stress during their service [1], and most are exposed to potentially traumatic stressors [2]. While many who have symptoms of PTSD appear to recover in the months following return from deployment [3], chronic Posttraumatic Stress Disorder (PTSD) and depression trouble a substantial number of veterans [4]. Prevalence rates for PTSD in military personnel and veterans who were deployed to Iraq or Afghanistan have been estimated to range from 5 to 20% [5], and rates of depression have been estimated to range from 5 to 37% [6]. In addition, from 1997 to 2005, the number of veterans with a diagnosis of PTSD receiving Veterans Administration (VA) specialty mental health services doubled and 87% were veterans who served in the Gulf War or earlier [7]. Given that delays in initiating treatment contributes to the chronicity of mental health problems in veterans [8], health care systems in both the Department of Defense (DoD) and VA have implemented screening programs aimed at early detection and treatment of mental health problems.

### Diagnostic screening vs risk screening methods

Although both the military and VA screen for common mental health conditions in primary care [9, 10], only a minority of veterans who screen positive receive adequate mental health care [11, 12]. In one study of veterans screened in a primary care system that integrated behavioral health care with primary medical care and employed a proactive referral system, less than half of veterans who screened positive for depression and/or PTSD attended mental health intake sessions or received any psychotherapy in the following 18 months [13]. Poor predictive performance of mental health screens and disconnection between perceived needs and the focus of screening may both be contributing to this lack of engagement in mental health care.

A systematic review of research on post-deployment mental health screening that uses a diagnostic model has found high variability in sensitivity of screens (.46 to .83) and low to moderate treatment initiation rates following screening [10]. This means that positive diagnostic screens are not consistently accurate in identifying those in need of care. Many with positive screens may recover on their own without mental health care. About 50% of those who screen positive for PTSD, major depression, alcohol misuse, or other mental health problems when they return from deployment to a war zone appear to recover without treatment within three to six months [3]. Thus, depending on the timing and circumstances of screening, positive diagnostic screening for current mental health problems may not be a good indicator of risk for persisting mental health problems.

Another contributor to veterans disinclination to accept mental health services may be that information given to them about their current status in of a mental illness diagnosis does not seem relevant or actionable. Among many systemic and individual factors that can be barriers to care [14–20], veterans' reluctance to engage in mental health care may be the greatest.

Qualitative and quantitative research has found that many veterans expect their problems to resolve without treatment, have a strong desire to solve problems without help, and hold beliefs that discourage engagement in care [21, 22]. Specifically, many have general attitudes of mistrust of others and negative attitudes about mental illness diagnostic labels, about standard models of mental health care offered by VA and about those who develop mental health problems and those who seek care [12, 16, 22, 23]. Information about current symptoms of a mental illness diagnosis may not be sufficient to overcome these barriers and persuade some veterans that there is a pressing need to engage in mental health care.

Screening veterans for risk of future mental health problems may address predictive accuracy and relevance problems associated with diagnostic screening. Information about risk for later mental health problems may seem more relevant and actionable to veterans than current diagnostic information. Those at risk could be told that veterans with risk screen scores like theirs did not recover on their own. Research has shown that risk information about outcomes of concern to patients can play a critical role in determining health-related behaviors. For example, in a large RCT of primary care patients, participants who completed a screen for cancer risk and received generic risk information or information about their personal colorectal cancer risk subsequently completed colorectal cancer medical tests at three times the rate of those who did not complete a risk screen and received no risk information [24]. Our success in developing a multi-risk mental health screening tool for hospital patients that performed well to predict later mental health problems [25] encouraged us to pursue a similar approach for veterans who were primary care patients.

## Theory and research on risk factors for posttraumatic and postdeployment mental health problems in veterans

This study was based on theoretical frameworks for the impact of traumatic stress [26, 27] and deployment stress [28] and research on risk factors for PTSD and depression following traumatic stress [29]. The etiologies of post-traumatic and post-deployment mental health problems are complicated because of overlap in exposure to trauma and deployment and because a large number of factors affect people's responses to both types of experiences. These factors include characteristics of individuals (such as genetic or biological tendencies, developmental level and experiences, past trauma exposure, life stress, and gender), characteristics of stressors (such as severity, intentional or accidental nature, and duration) and time-of-trauma factors (such as social support), and posttraumatic life experiences, responses, and resources (such as social support, availability and use of medical and psychological treatment, and posttraumatic life stress) [26, 27, 29]. These factors may influence responses to potentially traumatic stressors by affecting perceptions of stressful events, expectations about the impact of events, the capacity to cope with strong emotions related to events, or some combination of these.

Military personnel and veterans are subject to the same risk factors as those exposed to other types of traumatic stressors, and also to risks associated with serving in a war zone or deployment to other assignments inside or outside the country. Most research on veterans has focused on combat-related experiences, but other aspects of deployment may also influence health and mental health outcomes [28, 30]. These include perceived threat, aftermath of battle, difficult living and working environment, sense of preparedness, exposure to toxic agents, concerns about life and family disruptions, sexual or general harassment, social support during and after deployment, and life stress following deployment. Moral injury has also been identified as deleterious to mental health in veterans [31–35]. Moral injury has been defined as psychological damage caused by transgressions of deeply held moral or ethical beliefs [33]. More recently, moral injury has been defined as disturbances in virtues, character, and identity that

follow a moral failure event, and it has been posited that the pathways of influence of moral injury and traumatic stress are independent [36]. Because traumatic stress and moral injury events often co-occur, the experience of both may be especially damaging. To date, moral injury research has focused on veterans who were treatment-seeking, so little is known about the incidence of moral injury in other populations or its value as a prospective predictor of later mental health problems.

The quality of relationships with partners, family members, friends, and co-workers during and after military deployment can also influence mental health. Military personnel deployed in recent years often served longer in war zones and have been deployed for multiple tours more often than prior military cohorts [11]. These extended periods of time apart and experiences of war zone stress and traumatic stress make the transition from deployments to the homefront stressful. Veterans of Iraq and Afghanistan have reported high levels of stress in marital, family, and work relationships during and after military service [37, 38], but there has been little examination of the predictive value of reports of stress in relationships related to deployment or transition to civilian life and later mental health.

## Development and cross-validation of a veterans mental health risk screen in current project

In the current project, we sought to create a screen for risk of later mental health problems in veterans seeking primary care in a VA clinic and to perform cross-validation by examining the predictive performance of selected risk factor items in a new sample [39]. We focused on risk factors that are thought to play causal roles in the development of mental health problems in order to identify factors that are clinically relevant and modifiable. In the Screen Development Study, we conducted secondary analyses of data from prior research to identify which risk factors are most strongly related to PTSD and/or depression and to select small subsets of items to measure these risk factors. In the Cross-Validation Study, we collected data in a new sample of VA primary care patients (VA Primary Care II) on the selected risk factor items and on PTSD and/or depression six months later, and we calculated the performance of the risk screen results to prospectively predict these mental health problems [40, 41].

## Screen development study: Introduction

We sought to identify a set of risk factors that are strongly related to PTSD and depression by conducting secondary analyses of data from published studies of risk factors and mental health problems and new data on a novel risk factor and mental health status in VA primary care patients. For each of the most strongly-related risk factors, we then identified subsets of items to measure the risk factor by examining which items were most strongly related to total scores on the full risk measures, conducting factor analyses to determine which items represented the underlying constructs well, and conducting forward regressions to determine which items collectively accounted for at least 90% of the variance in total scores.

We chose datasets that were diverse in terms of time since exposure to trauma or deployment because veterans who seek VA primary care are similarly diverse in regard to the time that has passed since their exposure to deployment or traumatic stress. When available, we included datasets that included prospectively collected data on variables that were strongly related to later mental health outcomes. To examine risk factors that would relate to long-term mental health sequelae of traumatic and deployment stress in a broad sample of veterans that was not limited to those seeking mental health care, we included data from a diverse and nationally representative group of Gulf War veterans who had been deployed ten years earlier [28]. To examine risk factors that would relate to short-term mental health sequelae of

traumatic and deployment stress in veterans who were not all seeking health or mental health care, we included a dataset that was comprised of military personnel who served in Iraq or Afghanistan and were assessed upon return from deployment and three to six months later [42]. To examine risk factors related to mental health sequelae of traumatic stress that were not available in a military or veteran sample, we included a dataset that was comprised of patients and family members exposed to traumatic injury who were assessed within two weeks of trauma and two months later [25]. To examine the associations between symptoms and relationships related to deployment and transition to civilian life following deployment, we collected new data in a sample of veterans seeking VA primary care services. We focused on symptoms of PTSD, which are common among veterans receiving VA mental health care [43].

## Screen development study: Methods

The Administrative Panel on Human Subjects in Medical Research of Stanford University, the designated IRB for VA Palo Alto Health Care System research, approved this study as protocol 17192. For secondary datasets (Gulf War Veterans, Iraq and Afghanistan Military Personnel, Residential Veterans, VA Residential Veterans VA Homeless Domiciliary Residents, and Traumatic Injury Survivors) deidentified data were studied and consent was obtained from participants at the time the original research was conducted. For new data collected for the VA Primary Care I and VA Primary Care II samples, oral informed consent was obtained by participants.

### Gulf War veterans participants, procedures and measures

Data from Gulf War veterans were collected as part of a large national study that had the primary aim to develop a measure of deployment risk and resilience [44]. Potential participants were identified through records held by the Defense Manpower Data Center (DMDC), a central repository for Department of Defense personnel data and the VA Gulf War Health Registry. These veterans were deployed to the 1990–1991 Gulf War conflict (Gulf War I), and the data used were collected approximately ten years after deployment. Complete data were obtained from 317 (64%) of 495 who were sent questionnaires by mail. A detailed description of the study methods is provided in King (2006). Risk and resilience factors were assessed using the Deployment Risk and Resilience Inventory (DRRI) [44]. The DRRI is an collection of 14 scales for assessing predeployment, deployment, and postdeployment psychosocial factors that put veterans at risk for poor post-deployment health and adjustment. Posttraumatic stress symptoms were assessed with the military PTSD Checklist (PCL-M) [45].

### Iraq and Afghanistan (IA) military personnel participants, procedures and measures

Data from military personnel recently returned from deployment to Iraq or Afghanistan were collected as part of a study of mental and physical health status and health services use [42]. Potential participants were identified through the DMDC. A sample of 2,000 / (OEF/OIF) personnel who had returned from deployment to Iraq (Operation Iraqi Freedom) or Afghanistan (Operation Enduring Freedom) between 3 and 12 months prior was stratified by service component (50% Active, 25% National Guard, and 25% other Reserve) and gender (>50% women within each service). Of 1,833 eligible individuals, 1,043 (57%) received the initial survey, and 596 (57%) returned completed initial (T1) surveys in the time frame allowed. Survey respondents came from every state except Wyoming and from the District of Columbia, Puerto Rico, and the US Virgin Islands. Follow-up data (T2) were collected from 512 veterans (86% of the 596 who completed the T1 survey) between six and nine months following receipt of the T1

survey [46]. Risk and resilience factors were assessed at T1 with the DRRI [44] (described above). Symptoms at T1 and T2 were assessed with the PCL-M (described above) and the Behavior and Symptom Identification Scale (BASIS-24) [47]. The BASIS-24 uses 24 items to assess six symptom domains: depression/functioning, interpersonal relationships, psychotic symptoms, alcohol/drug use, emotional lability, and self-harm.

## Traumatic injury survivors participants, procedures and measures

Data from traumatic injury survivors were collected as part of research on psychological responses in patients hospitalized after severe, sudden injury and family members of severely injured patients [25]. Most risk factors, including early symptoms, were assessed upon enrollment between one and fourteen days after injury in 147 participants (54% patients and 46% family members). Social support and social constraints were assessed between eight and twenty-one days after injury in 64 participants. PTSD symptoms assessed two months post-injury were available for 129 participants. There were no significant differences between patients and family members on any risk factor or any outcome. PTSD was assessed with the Screen for Posttraumatic Stress Symptoms (SPTSS) [48, 49]. In psychiatric inpatients, scores on the SPTSS showed strong internal consistency ($\alpha = 0.91$) and good concurrent validity when correlated with other measures of PTSD [48]. In a study of recent combat veterans with a 15% rate of PTSD, scores on the SPTSS had a sensitivity of 0.89 and specificity of 0.89 to predict PTSD diagnosis from the SCID PTSD module [49].

## VA residential veterans participants, procedures and measures

Data were collected from 240 Vietnam War veterans participating in a U.S. Veterans Health Administration residential treatment program for chronic and severe PTSD related to combat or other trauma exposure during military service. Data were collected at the time of admission as part of research on the psychometric properties of a measure of dissociation [50]. Data included in this study were scores on the PCL-M (described above) and the Dissociative Symptoms Scale (DSS). The DSS assesses dissociative symptoms, which are distortions in perceptions, attention, and memory that are strongly related to exposure to traumatic stress and to PTSD symptoms [50]. Scores on the DSS in this sample showed strong reliability and validity in Carlson et al. [50].

## VA homeless domiciliary residents participants, procedures and measures

Data were collected from 115 veterans participating in a U.S. Veterans Health Administration residential program for homeless veterans. Data were collected as part of research on military and civilian trauma exposure and posttraumatic symptoms [51]. Data included in this study were scores on the PCL-M and the DSS (both described above). Findings of strong reliability and validity for scores on the DSS in this sample are reported in Carlson et al. [50].

## VA primary care (I) participants, procedures, and measures

Data were collected from 131 VA primary care patients at a VA medical center. In order to limit the potential for errors in reports of past experiences due to the passage of time, inclusion was limited to veterans who were within five years of military discharge. Appointments were identified in the electronic medical records for primary care, laboratory services, audiology, dental, optometry, orthopedics, gastro-intestinal, and hand and upper extremity. While in the waiting area, veterans who did not opt out were invited to participate and provided informed consent. Data reported here included demographics, employment, and military service.

Stress in relationships related to deployment experiences or transition to civilian life was assessed in the 118 veterans who had been deployed during military service using a measure of stress in relationships called the Deployment Transitions Stress Scale (DTSS) that was developed to quantify stress in marital/partner and family relationships related to deployment and return from deployment and about changes in relationships with friends and in the work place since deployment. The DTSS was based on the Transitioning Families Questionnaire, an unpublished clinical interview authored by Erika Curran that was used to collect relationship information from VA residential PTSD treatment program patients and their spouses. The DTSS includes items focused on stress in the relationship with a spouse or partner, family members, friends, and work. Example items include "Compared to before deployment, how many disagreements do you and your partner have now? (fewer, about the same, somewhat more, a lot more); "Since returning, how hard is it to talk or answer questions about your war experiences with your children or other family members?" (not at all hard, a little bit hard, somewhat hard, very hard); "How much have you felt alienated from your friends since returning from deployment?" (not at all, a little bit, somewhat, very much); "How hard has it been for you to get along with coworkers in a new workplace since returning from deployment?" (not at all, a little bit, somewhat, very hard). Data on Relationship Stress: Spouse/Partner were available for the 32 veterans who were still in relationships with the same spouse/partner as at the time of deployment. Data on Relationship Stress: Friends were available for N = 104 of 118 who had deployed during military service. Cronbach's alpha value for the total DTSS was 0.93 and values for the subscales were 0.76 (spouse/partner), 0.64 (family); 0.64 (friends), and 0.82 (work).

The SPTSS (described above) was used to assess DSM-IV PTSD symptoms.

## Data analysis

We used common methods for shortening scales [52–54] to select subsets of items with the goal of creating very brief risk factor measures that could predict mental health outcomes. Creating short forms of hypothetical constructs that produce scores with strong psychometric properties typically involves shortening measures of 30 or more items to about 12 items [53]. However, a screen made up of a collection of typical short forms would be too long to use for screening in health care settings. Further, since some of the most predictive risk factors (e.g., PTSD symptoms) are multifactorial [55], combining short forms that assessed all of the underlying factors of these multi-factor risks would likewise produce a multi-risk screen that was too long for practical use. For these reasons, we did not seek to create brief measures of the entire content domains of risks or produce scores with psychometric properties strong enough to serve as independent measures of the constructs. Instead, we sought to identify a set of risks that could be assessed with relatively few items and demonstrate good predictive validity of future mental health status.

Some risk factor variables we studied are hypothetical constructs (e.g., social support), whereas others are composite measures made up of items that represent discrete, possibly uncorrelated experiences (e.g., combat experiences) [56]. To select items for brief measures of risk factors that are hypothetical constructs, we examined item-total correlations and favored items that were most highly correlated with total scores [52]. We also conducted factor analyses to examine whether risk measures included only one or multiple factors, and eliminated lower-loading items by factor. To shorten measures of risk factors that are composite variables, we relied primarily on removing items that related less strongly to total scores. For all risk factors (constructs and sets of experiences), in order to maximize criterion-related validity with respect to the original full scales, we conducted forward regressions with items as predictors of

the score for the full risk factor measure and retained enough items to account for at least 90% of the variance in the full measure score. We used data from the IA Military sample to select items for brief measures of some deployment-related variables rather than data from the Gulf War sample because the former was a much larger sample and the data were collected prospectively and over a comparable time frame between risks and PTSD symptoms to be predicted.

## Screen development study: Results and discussion

The number of participants, demographic characteristics, and service information for the samples studied are shown in Table 1. In the VA Primary Care (I) sample, the mean time since return from most recent deployment was 27 months (sd = 20.5). Table 2 shows the risk factors and symptoms assessed in each study, the measures used to assess them, and correlations between risk factors and outcomes in each dataset.

The number of items selected for each risk factor and the correlation between total scores on brief and full measures of constructs in the Screen Development Study are shown in Table 3. These correlations were generally quite high with most over $r = .85$, but were not as high as $r = .95$ suggested for creation of short forms [57].

**Table 1. Characteristics of the samples.**

| | Gulf War (N = 317) | IA[a] Military (N = 512) | Injury (N = 129) | VA PTSD Resident (N = 240) | VA Homeless (N = 115) | VA Primary Care I (N = 131) | VA Primary Care II[b] (N = 232) |
|---|---|---|---|---|---|---|---|
| Gender: Male | 74% | 40% | 42% | 89% | 96% | 86% | 85% |
| Age: Mean (SD) | 44 (9.0) | 34 (9.2) | 44 (14.1) | 52 (5.6) | 45 (6.3) | 33 (9.5) | 44 (16.2) |
| Hispanic Ethnicity | 14% | 13% | 18% | 13% | 4% | 17% | 23% |
| Race | | | | | | | |
| White | 74% | 76% | 74% | 57% | 46% | 66% | 62% |
| Black | 15% | 17% | 4% | 22% | 47% | 7% | 13% |
| Other or multiple | 11% | 7% | 22% | 8% | 3% | 27% | 35% |
| Marital Status | | | | | | | |
| Single | 7% | 21% | 20% | 7% | 35% | 46% | 35% |
| Married/Partner | 74% | 67% | 56% | 33% | 4% | 26% | 40% |
| Separated/ Divorced | 18% | 12% | 24% | 56% | 57% | 28% | 24% |
| Education | | | | | | | |
| High school/GED | 7% | 18% | 20% | - | - | 15% | - |
| Some college/voc ed | 52% | 51% | 31% | - | - | 55% | - |
| Bachelors or more | 41% | 31% | 47% | - | - | 30% | - |
| Employment | - | 83% | - | - | - | 60% | - |
| Service Type | | | | | | | |
| Active | 26% | 80% | - | - | - | 70% | 88% |
| National Guard/ Reserve | 74% | 20% | - | - | - | 30% | 12% |
| Deployed | 100% | 100% | - | - | - | 85% | 81% |

Notes

[a]IA = Iraq and Afghanistan

[b]VA Primary Care II sample was collected as part of the Cross-Validation Study.

**Table 2. Risk factors assessed, measures used, and correlations between risk factors and PTSD symptoms in six samples.**

| Risk Factor | Measure | Gulf War Vets (N = 317) | IA Military Post-Deploy (N = 512) | Traumatic Injury (N = 129) | VA PTSD Residential (N = 240) | VA Homeless Domiciliary (N = 115) | VA Primary Care I (N = 104) |
|---|---|---|---|---|---|---|---|
| Pre-trauma Individual Characteristics | | | | | | | |
| Education | - | -.31 | -.25 | | | | |
| Prior Stressors | DRRI-A | | .37 | | | | |
| Past High Magnitude Stressors | THS | | | .35 | | | |
| Childhood Family Environment | DRRI-B | -.06[a] | -.25 | | | | |
| Childhood Home Life | - | | | -.34 | | | |
| Preparedness for Deployment | DRRI-C | -.23 | -.34 | | | | |
| Perceived Life Stress (Pre-trauma) | PSS | | | .46 | | | |
| Stressor Characteristics and Time-of-Trauma Factors | | | | | | | |
| Difficult Living and Working Environment | DRRI-D | .54[a] | .48 | | | | |
| Concerns about Life and Family Disruptions | DRRI-E | .46[a] | .40 | | | | |
| Deployment Social Support | DRRI-F | -.34[a] | -.38 | | | | |
| General Harassment | DRRI-G1 | .44[a] | .40 | | | | |
| Sexual Harassment | DRRI-G2 | .29[a] | .32 | | | | |
| Perceived Threat | DRRI-H | .53[a] | .53 | | | | |
| Combat Experiences | DRRI-I | .40[a] | .45 | | | | |
| Aftermath of Battle | DRRI-J | .43[a] | .44 | | | | |
| Nuclear, Biological, & Chemical Exposure | DRRI-K | .45[a] | .35 | | | | |
| Length of time in the military | | -.14[a] | | | | | |
| Applied for service disability | | .43[a] | | | | | |
| Post-Trauma Experiences, Responses and Resources | | | | | | | |
| Post-deployment Social Support | DRRI-L | -.22[a] | -.53 | | | | |
| Post-deployment Stressors | DRRI-M | .41[a] | .33 | | | | |
| Relationship Stress: Spouse | DRRS | | | | | | .80[a, b] |
| Relationship Stress: Friends | DRRS | | | | | | .73[a, c] |
| Perceived Life Stress (Post-trauma) | PSS | | | .70 | | | |
| Social Constraints | SCS | | | .59[d] | | | |
| Social Support | SSS | | | -.42[d] | | | |
| General Self-Efficacy | GSES | | -.39 | | | | |
| PTSD Symptoms | PCL-M | | .79 | | | | |
| PTSD Symptoms | SPTSS | | | .62 | | | |
| Dissociation | DSS | | | .52[a] | .46[a] | .73[a] | |
| Negative Thinking | PTCI | | | .57 | | | |
| Depression | BDI-SF | | | .50 | | | |
| Depression/Anxiety score | BASIS-R | | .61 | | | | |

*(Continued)*

**Table 2.** (Continued)

| Risk Factor | Measure | Gulf War Vets (N = 317) | IA Military Post-Deploy (N = 512) | Traumatic Injury (N = 129) | VA PTSD Residential (N = 240) | VA Homeless Domiciliary (N = 115) | VA Primary Care I (N = 104) |
|---|---|---|---|---|---|---|---|
| Psychosis (Interpersonal Threat) | BASIS-R | | .55 | | | | |
| Emotional Lability | BASIS-R | | .52 | | | | |
| Interpersonal Problems | BASIS-R | | .45 | | | | |
| Substance Abuse | BASIS-R | | .31 | | | | |
| Days in past month with medical problems | | | .40 | | | | |
| Alcohol Use | AUDIT | .21 | | | | | |

Notes

[a]Risk factor and outcome data collected at same time point.

[b]Data on Relationship Stress: Spouse/Partner were available for N = 32 who were still in relationships with the same spouse/partner as at the time of deployment.

[c]Data on Relationship Stress: Friends was available for N = 104 of 118 who had deployed during military service.

[d]Data on post-trauma Social Constraints and Social Support were available for N = 64 participants. AUDIT = Alcohol Use Disorders Test, BASIS-R = Behavior Symptom Identification Scale–Revised, BDI-SF = Beck Depression Inventory–Short Form, DRRI = Deployment Risk and Resilience Inventory, DSS = Dissociative Symptoms Scale, DTSS = Deployment and Transitions Stress Scale, GSES = General Self-Efficacy Scale, IA = Iraq and Afghanistan, PCL-M = Posttraumatic Stress Checklist–Military, PSS = Perceived Stress Scale, PTCI = Posttraumatic Cognitions Inventory, SCS = Social Constraints Scale, SSS = Social Support Survey, SPTSS = Screen for Posttraumatic Stress Symptoms, THS = Trauma History Screen

**Table 3. Number of Items in brief measures and correlations between brief and full measures.**

| Risk Factor | Number of Items | IA Military Post-Deploy (N = 512) | VA PTSD Resident & Homeless (N = 355) | Traumatic Injury (N = 129) | VA Primary Care-I (N = 131) |
|---|---|---|---|---|---|
| PTSD Symptoms | 5 | | | | .95 |
| Deployment & Transition Stress: Spouse/Partner | 2 | | | | .86[a] |
| Deployment & Transition Stress: Friends | 1 | | | | .80 |
| Depression | 3 | .94 | | | |
| Social Constraints | 2 | | | .90 | |
| Negative Thinking | 4 | | | .91 | |
| Interpersonal Threat | 1 | .86 | | | |
| Deployment Environment | 4 | .86 | | | |
| Deployment Concerns | 3 | .88 | | | |
| Dissociation | 5 | | .96 | | |
| Emotional Lability | 1 | .87 | | | |
| Life & Family Concerns | 5 | .85 | | | |
| Perceived Life Stress | 2 | | | .92 | |

Notes

[a]Data on Deployment and Relationship Stress: Spouse/Partner available for N = 32 veterans.

[b]Data for VA PTSD Resident and Homeless samples were combined for this analysis because the veterans in the samples are similar and using a larger sample is better to select items for the brief measure of dissociation. IA = Iraq and Afghanistan.

## Cross-validation study: Introduction

In the Cross-validation Study, we recruited a new sample of VA primary care patients (VA Primary Care II) to study the predictive performance of risk factor items selected in the Screen Development Study on symptoms of PTSD and depression six months later. We included symptoms of depression because they are so common among veterans receiving VA mental health care [43]. We used elevation in symptoms rather than diagnosis as the criterion, because diagnosis of mental disorder is highly stigmatizing [58], and effective implementation of a risk screen would require a more patient-centered approach. Use of self-report measures rather than diagnostic interviews also results in study samples that more closely reflect the intended population because a more representative sample can be enrolled and a higher proportion are retained than if a long diagnostic interview were required.

Given that predicting future mental health status is the goal of the screen, we examined the predictive or prospective validity of multiple risk factors to predict later mental health [52]. We collected data on the risk factors listed in Table 3 and on moral injury from VA primary care patients and data on symptoms of PTSD and depression six months later. We conducted analyses to: 1. Confirm that, in the new sample, the risk factors assessed by brief measures related strongly to PTSD and depression symptoms six months later. 2. Determine the subset of risk factors that most strongly predicted PTSD and depression status six months later. 3. Examine the performance of that set of risk factors to prospectively predict later elevated PTSD and/or depression symptoms. 4. Examine the predictive performance of the subset of risk factors in the subsample of veterans who identify as belonging to an ethnic or racial minority identity.

## Cross-validation study: Methods

### Participants and procedures

Patients were recruited for the cross-validation study using the same methods and procedures as described above in the VA Primary Care I sample. At the time of enrollment, participants completed risk factor measures. Six months following enrollment, participants completed mental health measures and reported on major life stressors since enrollment.

### Measures

The risk factors identified in the Screen Development Study (shown in Table 3) were assessed with the items selected in the Screen Development Study. To make them applicable to all veterans, simplify wording, and combine very similar constructs, some items were modified. For example, items originally focused on deployment were reworded to "During military service," and items originally specifying spouses or partners were modified to apply to those who did not have partners or spouses (e.g., refer to "your relationships" instead of "partner or spouse"). For all items, a simple, uniform set of response options was used of "not at all", "a little bit", "some", "a lot" with risk factor scores coded to have higher scores reflect less favorable experiences or higher levels of symptoms.

Moral injury was also assessed with four novel items developed in consultation with four experts on moral injury. All members of the expert panel had doctoral level training in psychiatry or psychology and lived experience of military service and had published and presented on the concept of moral injury. After discussion about the construct of moral injury, four items were developed to assess the construct: "been bothered about things that happened during military service that were just not right"; "felt bad about yourself because of things you saw or did during military service"; "felt guilty about things you did or didn't do while in the

military"; and "been bothered by killing people during military service". When administered, items were preceded by "In the past month, how much have you. . ." and response options were "not at all", "a little bit", "some", "a lot".

The SPTSS (described above) was used to assess PTSD symptoms six months after enrollment. SPTSS scores of 20 or higher were used to categorize participants as having elevated PTSD symptoms. This was based on strong performance of scores on the SPTSS to predict PTSD diagnosis by the Structured Clinical Interview for DSM-IV (SCID) in a sample of veterans (N = 317; SE = .89; SP = .89) [49] and by performance of scores of 20 or more on the SPTSS to predict PTSD diagnosis on the Clinician-Administered PTSD Scale (CAPS) in a sample of adults assessed two months after traumatic injury of themselves or a close family member (N = 40, SE = .90, SP = .80) [40].

The Patient Health Questionnaire—9 (PHQ-9) was used to assess depression symptoms over the past two weeks. The PHQ-9 has 9 items with 4 response options that range from "not at all" to "nearly every day". The PHQ-9 has strong internal consistency (α = .89), test-retest reliability (r = .84), and correlations with other measures of depression severity and excellent SE (.95) and SP (.84) to a depression diagnosis [59, 60]. PHQ-9 scores of 10 or more were used to categorize participants as having elevated depression based on Veterans Health Administration/ Department of Defense guidelines for assessment of depression in primary care [61].

To identify veterans who experienced new traumatic stressors between the time of enrollment and the follow-up assessment six months later, the follow-up assessment included the question "Since you completed the first part of this study (about 6 months ago), have you had any major life events or new major stressors?" and descriptions of the events/stressors were collected for those who answered "yes". These descriptions were used to identify veterans who had been exposed to potentially traumatic stressors.

## Data analysis

Potentially traumatic stressors and multiple major stressors experienced in the six months following enrollment were expected to increase mental health symptoms and affect accuracy of prediction for veterans expected to have no disorder at six months. Therefore, data were excluded from analyses for four participants whose screen results predicted no disorder at six months and who reported major stressors between enrollment and follow-up that were considered potentially traumatic. All analyses reported here were conducted on the 228 participants who completed the follow-up and reported no new major life events or stressors that were considered likely to be traumatic stressors.

The sum of standardized scores on the SPTSS and the PHQ-9 was used as an index of mental health that reflected symptoms of *both* PTSD and depression. To determine the subset of risk factors that most strongly predicted elevation in combined PTSD and depression symptoms six months later, we conducted a forward regression to predict the index of mental health. The statistical software analyzed the variance in the outcome (index of mental health six months after screening) associated with all risk factors listed in Table 3 and added risk factors to the model in a stepwise fashion. At each step, the risk factor was added that provides the best improvement to the model and contributes significant additional variance with set at $p < .05$. We conducted a Receiver Operating Characteristic (ROC) analysis to identify a cut score on the subset of risk factors that maximized SE to predict mental health status without letting SP fall below 0.70. We examined the performance of the cut score to predict elevated mental health symptoms six months later.

**Table 4. Correlations in cross-validation study between scores for selected risk factor item sets at enrollment and PTSD and depression symptoms 6 month later.**

| Risk Factor | Number of Items | Cronbach's alpha | r with PTSD (N = 228) | r with depression (N = 228) |
|---|---|---|---|---|
| PTSD Symptoms | 5 | .89 | .73*** | .64*** |
| Relationship Stress | 3 | - | .62*** | .56*** |
| Depression | 3 | .83 | .67*** | .64*** |
| Social Constraints | 2 | .79 | .68*** | .59*** |
| Negative Thinking | 4 | .88 | .73*** | .66*** |
| Interpersonal Threat | 1 | - | .56*** | .54*** |
| Difficult Living and Working Environment | 4 | - | .23*** | .14* |
| Perceived Threat | 3 | - | .45*** | .34*** |
| Dissociation | 5 | .87 | .77*** | .66*** |
| Emotional Lability | 1 | - | .67*** | .64*** |
| Concerns about Life and Family Disruptions | 5 | - | .38*** | .37*** |
| Perceived Life Stress | 2 | .82 | .69*** | .66*** |
| Moral Injury | 4 | .84 | .62*** | .56*** |

Note

*** = $p < .0001$

* = $p < .05$

## Cross-validation study: Results

A sample of 284 veterans were enrolled and completed the risk measures. Symptoms measures were completed six months later by 232 (82%) of those enrolled. The demographic characteristics and service information for the cross-validation sample (VA Primary Care II) are shown in Table 1. The median length of time enrolled for health care at the VA facility which was the study site was five months; 65% had enrolled within the past year, and 82% had enrolled within the past two years.

Table 4 shows the number of items in brief measures of each risk factor, Cronbach's alpha values in the new sample for brief risk factors that are constructs, and correlations between risk factors measured at the time of enrollment and PTSD and depression symptoms assessed six months later. As noted above, we only included data from those who reported no new potentially traumatic stressors during the follow-up period. While PTSD and depression assessed six months after risk factors were highly related ($r = .83$), the risks most strongly related to each were not the same.

A forward regression to predict the combined index of PTSD and depression six months after enrollment yielded $R = .80$ ($df = 4, 208$; $p < .001$) with risk factors of dissociation, emotional lability, life stress, and moral injury in the model. Table 5 shows the classification performance of these four risk factors assessed by 12 items to predict elevations in the combined index of PTSD and depression, in PTSD, and in depression. In patients reporting an ethnic/racial minority identity (n = 110), performance of a screen with 5 risks (17 items) to predict elevated PTSD and/or depression correctly classified 76% with SE = .90, SP = .66.

**Table 5. Classification performance of screen including four risk factors to predict elevations in a combined index of PTSD and depression, PTSD, and depression.**

| Risks Assessed (items per risk) | Criterion | Positive if ≥ | % Accurately Classified | SE | SP |
|---|---|---|---|---|---|
| Dissociation (5) | Elevation Combined PTSD and Depression | 9 | 79% | .86 | .75 |
| Emotional Lability (1) | Elevation PTSD Symptoms | 9 | 78% | .93 | .70 |
| Life Stress (2) | Elevation Depression Symptoms | 9 | 76% | .86 | .71 |
| Moral Injury (4) | | | | | |

## Cross-validation study: Discussion

This two-part study was designed to create a screen for risk of later mental health problems in veterans seeking primary care in VA. Analyses identified risk factors that were strongly related to PTSD symptoms in samples of veterans, military personnel recently returned from deployment, adults recently exposed to traumatic injury, veterans in residential treatment programs, and veteran primary care patients. Items were selected to assess the most predictive risk factors, and data on risks and mental health six months later were collected for a new sample of veterans. Twelve items assessing dissociation, emotional lability, life stress, and moral injury showed strong predictive validity with high sensitivity and specificity for later elevated levels of PTSD and depression. Sensitivity was also high for veterans who identified as belonging to one or more ethnic or racial groups, but specificity was lower.

Performance of this screen was comparable to that of similar mental health risk screens. Total scores for the 18 items assessing four risks on the Hospital Mental Health Risk Screen showed a sensitivity of .86 and a specificity of .72 [40]. Total scores for the 18 items assessing 4 risks on the Military Mental Health Risk Screen showed a sensitivity of .80 and a specificity of .86 [41]. Availability of this accurate screen for mental health risk is only the first step to increase engagement of at-risk veterans in mental health care. Further research is needed on the impact of mental health risk information in combination with readily accessible and appealing options to reduce risk.

A strength of the study is that data from five samples of military veterans were analyzed to select the most predictive risk factors to study. Recommended psychometric methods were used to select subsets of items to measure the risk factors, and the brief risk factor measures demonstrated strong internal consistency in a new sample. Inclusion of moral injury among the risk factors studied made it possible to determine that this difficult type of experience [31] makes a unique predictive contribution. While variables such as PTSD, depression, negative thinking and social constraints at the time of screening were more strongly related to PTSD six months later, those variables covaried with risk factors already in the model (dissociation, emotional lability, and life stress) whereas moral injury accounted for unique variance. It is also a major strength that the capacity of the risk factors to prospectively predict later mental health was studied in a new sample. Lastly, we were able to recruit a very diverse sample and to retain 82% of participants, which allowed evaluation of screen performance in patients identifying with one or more ethnic or racial minority identities.

A limitation of the study is that insufficient data were collected to select brief measure items using Item Response Theory methods, which are more advanced methods that indicate the amount of information about underlying constructs provided by items [62]. However, measures developed based on classical test theory are often very similar in quality to those developed based on IRT methods. Lastly, since many risk factors were strongly predictive of later PTSD and depression, many different sets of risk factors would accurately predict later mental health.

## Conclusions

Mental health status was prospectively predicted in veterans new to VA primary care with high accuracy using a screen that is brief, easy to administer, score, and interpret, clinically feasible in a variety of settings, and fits well into VA's primary care which integrates medical and mental health care [63]. These findings should be replicated to confirm the predictive validity of screen scores. Use of a predictive screen could foster prevention and early intervention research. Future research might investigate whether providing veterans with information about their mental health risk results in higher rates of engagement than providing

information about their current mental health status. Risk information seems likely to have the greatest impact when care is readily accessible, appealing to veterans, and not perceived as stigmatizing.

## Acknowledgments

The contents of this article do not represent the views of the US Department of Veterans Affairs or the US Government.

The authors wish to thank the following experts for their input during the design of this research: Captain Paul S. Hammer, M.D., Dr. Jonathan Shay, Colonel Carl Castro, Ph.D., Neil Greenberg, M.D., and William Nash, M.D.

## Author Contributions

**Conceptualization:** Eve B. Carlson, Patrick A. Palmieri, Dawne Vogt, Steven E. Lindley.

**Data curation:** Eve B. Carlson, Patrick A. Palmieri, Dawne Vogt, Kathryn Macia.

**Formal analysis:** Eve B. Carlson, Patrick A. Palmieri, Dawne Vogt, Kathryn Macia.

**Funding acquisition:** Eve B. Carlson, Patrick A. Palmieri, Dawne Vogt, Steven E. Lindley.

**Investigation:** Eve B. Carlson, Patrick A. Palmieri, Dawne Vogt, Kathryn Macia, Steven E. Lindley.

**Methodology:** Eve B. Carlson, Patrick A. Palmieri, Dawne Vogt, Steven E. Lindley.

**Project administration:** Eve B. Carlson, Patrick A. Palmieri, Kathryn Macia, Steven E. Lindley.

**Resources:** Eve B. Carlson.

**Supervision:** Eve B. Carlson.

**Validation:** Eve B. Carlson.

**Writing – original draft:** Eve B. Carlson, Patrick A. Palmieri, Dawne Vogt, Kathryn Macia, Steven E. Lindley.

**Writing – review & editing:** Eve B. Carlson, Patrick A. Palmieri, Dawne Vogt, Kathryn Macia, Steven E. Lindley.

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
