## [Decision Letter · Decision Letter 0]

18 Mar 2022

PONE-D-21-33139Development and cross-validation of a veterans mental health risk factor screenPLOS ONE

Dear Dr. Carlson,

Thank you for submitting your manuscript to PLOS ONE. After careful consideration, we feel that it has merit but does not fully meet PLOS ONE’s publication criteria as it currently stands. Therefore, we invite you to submit a revised version of the manuscript that addresses the points raised during the review process.

We look forward to receiving your revised manuscript.

Kind regards,

Renan C. Castillo

Academic Editor

PLOS ONE

Journal Requirements:

Additional Editor Comments (if provided):

I would like to commend you on an interesting and well executed study. The reviewers had only minor revisions for you and I agree with their recommendations. As you can see from their reviews, the major substantive item is the request for a better description/justification of the rationale for the tool and the development of these specific items. I would encourage the authors to pay particular attention to these items in their response.

Reviewers' comments:

Reviewer's Responses to Questions

**Comments to the Author**

1. Is the manuscript technically sound, and do the data support the conclusions?

Reviewer #1: Yes

Reviewer #2: Yes

2. Has the statistical analysis been performed appropriately and rigorously? 

Reviewer #1: Yes

Reviewer #2: Yes

3. Have the authors made all data underlying the findings in their manuscript fully available?

Reviewer #1: No

Reviewer #2: No

4. Is the manuscript presented in an intelligible fashion and written in standard English?

Reviewer #1: Yes

Reviewer #2: Yes

5. Review Comments to the Author

Reviewer #1: Thank you for your work on this manuscript and the opportunity to review it. On the whole, I considered the content interesting and your presentation very detailed. I recognize this was a lot to present in a single paper and therefore quite a bit of effort, and in that light, I would like to offer a few thoughts about the manuscript overall.

I thought the amount of detail was at times overwhelming, particularly leading up to the sections of the manuscript that spoke directly to the new work being presented. Overall, I recognize the importance in laying out this additional background as it is the foundation for the Screen Development and Cross Validation Studies, but I that some of the landscape and prior studies might have been more easily digested with the presentation of some of this information in either a tabular format or perhaps with a figure that laid out the overall flow of the work. This wasn’t a fatal flaw in my opinion, but I did feel the need to be extra careful in my reading to properly follow along.

Narrowing my comments to specific items in your manuscript, there are several points where additional clarification might be beneficial for the reader.

1. I’m not sure that I fully understood the motivation for developing this particular new screening method. Lines 141-145 indicate that you modeled the process for developing the current tool after those used for two other successful screening tools. My best understanding of the need for a new screener was to have something appropriate to the primary care clinic setting. If that is the case, I would recommend noting that more explicitly and/or what the improvement being sought here is.

2. Line 395 was the first time it became very clear about what the “new” work was in terms of the development and presentation of new data. Because I read the manuscript first, I did not carry an impression of what to expect through the reading of the manuscript that I realized later was more clearly laid out in the abstract. I thought that giving a little more information about what comprised the second half of the paper earlier in the text would have been helpful. As an example, I was first puzzled why the VA Primary Care-II cohort was not included in Tables 2 and 3 but it had been included with Table 1. Only after getting toward the end did I fully appreciate that this was the new cohort being described.

3. Line 48: It might be nice to include the range here to support the “high variability” noted here.

4. Line 54: References cited could be reflected as 12-18 unless there is a typographical error here.

5. Line 108-110: This sentence might benefit by adding a reference.

6. Lines 235-237, Tables 1, 2 and 3: There are 3 different numbers associated with the VA Primary Care-I cohort: 114 (line 235), 131 (table 1) and 104 (tables 2 and 3). It is not clear what these differences are or why they are presented with different denominators.

7. Lines 318-319: It appears this sentence is inadvertently cut short.

8. Line 332: One iteration of the word “identity” should be “identify.”

9. Line 380: Is the phrase, “who reported major stressors...” supposed to be, “who reported no major stressors...”?

10. Line 381-383: This sentence is important to understanding why Table 4 does not include 232 patients, so it may be better to locate it closer to the tables and the mention of 232 patients (line 395).

11. Table 3: Does not include the Gulf War cohort. Is there a reason for this?

12. Table 3: For clarity, it might be useful to note that the VA PTSD Resident & Homeless cohort is the combination of two cohorts presented separately in Tables 1 and 2? Is there a reason these are combined here but not elsewhere?

13. Table 4: Dissociation is listed as having 4 items here but having 5 items in tables 3 and 5.

14. Table 5: The caption notes 5 risk factors but only lists 4 in the table.

15. There is a mix of citation styles used in the manuscript that should be unified.

16. Abbreviations that are not specified in the text and tables include:

a. PTSD: Line 29

b. VA: Line 33

c. VHA: Line 36

d. SPTSS: Line 214

e. IA (presumably Iraq/Afghanistan): Tables 1, 2, and 3

Thank you again for your work on this manuscript.

Reviewer #2: This is a well-done and important study. I have only a few comments for the authors to consider.

1) The authors make the case that 'moral injury' is a uniquely predictive risk factor for PTSD and Depression and developed 4 novel items to assess moral injury. Because these items were developed de novo for this study, I would recommend that the authors further substantiate their decision to include these items instead of the 'next best' predictive risk factor, e.g., depression or social constraints. I also suggest a bit more description of the construct and/or noting the 4 assessment items developed by the panel since this construct is less well-understood than the other risk factors.

2) The authors note that a limitation of the study is that the population in the cross-validation study is proportionately younger than the population served by the VA in general. It would be helpful for the reader to know to what degree the authors expect that this difference would impact the external validity of their results, i.e., how big of a deal is this? Does this have implications for future studies to confirm the risk factor assessment?

3) On line 319, there appears to be a word missing from the end of the sentence, "...a more patient centered ____". I think this is missing the word "model" or "approach".

4) This line (323-324) is a worded a little confusingly: "We examined the predictive or prospective validity of multiple risk factors as this is the form of criterion-related validity that is relevant when predicting future status is the goal".

6. PLOS authors have the option to publish the peer review history of their article (what does this mean?). If published, this will include your full peer review and any attached files.

Reviewer #1: No

Reviewer #2: No

---

## [Author Response · Author response to Decision Letter 0]

17 May 2022

Additional Editor Comments:

I would like to commend you on an interesting and well executed study. The reviewers had only minor revisions for you and I agree with their recommendations. As you can see from their reviews, the major substantive item is the request for a better description/justification of the rationale for the tool and the development of these specific items. I would encourage the authors to pay particular attention to these items in their response.

***Thanks! Will do! Responses to Reviewer comments are marked by *** below.

Reviewers' comments:

Reviewer's Responses to Questions

Comments to the Author

3. Have the authors made all data underlying the findings in their manuscript fully available?

Reviewer #1: No

Reviewer #2: No

***We have changed our Data Availability Statement to address this issue:

Datasets for the Screen Development Study and the Cross-Validation Study used to calculate results reported in this publication will be shared via Dryad with identifying information removed. 

Reviewer #1: Thank you for your work on this manuscript and the opportunity to review it. On the whole, I considered the content interesting and your presentation very detailed. I recognize this was a lot to present in a single paper and therefore quite a bit of effort, and in that light, I would like to offer a few thoughts about the manuscript overall.

I thought the amount of detail was at times overwhelming, particularly leading up to the sections of the manuscript that spoke directly to the new work being presented. Overall, I recognize the importance in laying out this additional background as it is the foundation for the Screen Development and Cross Validation Studies, but I that some of the landscape and prior studies might have been more easily digested with the presentation of some of this information in either a tabular format or perhaps with a figure that laid out the overall flow of the work. This wasn’t a fatal flaw in my opinion, but I did feel the need to be extra careful in my reading to properly follow along.

***We appreciate that the amount of detail is a lot to process. Rather than adding one more table or figure, we thought it best to eliminate the paragraphs describing prior research and replace them with sentence at the end of the 4th paragraph of the introduction that mentions the development of multi-risk mental health risk screening tool for hospital patients that preceded this study. See lines 78-80 of the revised MS.***

Narrowing my comments to specific items in your manuscript, there are several points where additional clarification might be beneficial for the reader.

1. I’m not sure that I fully understood the motivation for developing this particular new screening method. Lines 141-145 indicate that you modeled the process for developing the current tool after those used for two other successful screening tools. My best understanding of the need for a new screener was to have something appropriate to the primary care clinic setting. If that is the case, I would recommend noting that more explicitly and/or what the improvement being sought here is.

***Sorry we didn’t make this point more effectively! We have rewritten the 2nd to 5th paragraphs of the introduction (***lines 38-80) to make the rationale clearer. The revised paragraphs emphasize that current diagnostic screening methods may fail to engage veterans into care because positive screens for current symptoms in veterans are a poor indicator of risk for persisting mental health problems and because information about current symptoms for a mental illness diagnosis may not seem relevant or actionable to veterans. Accurate information about risk for persisting mental health problems, on the other hand, may seem more relevant and actionable.*** 

2. Line 395 was the first time it became very clear about what the “new” work was in terms of the development and presentation of new data. Because I read the manuscript first, I did not carry an impression of what to expect through the reading of the manuscript that I realized later was more clearly laid out in the abstract. I thought that giving a little more information about what comprised the second half of the paper earlier in the text would have been helpful. As an example, I was first puzzled why the VA Primary Care-II cohort was not included in Tables 2 and 3 but it had been included with Table 1. Only after getting toward the end did I fully appreciate that this was the new cohort being described.

***Thanks for this suggestion. We have revised the last two sentences of the last paragraph of the introduction (lines 127-133) to clarify the study design and specify that the Cross-Validation Sample was the Primary Care II sample. We also clarified this in the notes for Table 1, on lines 318-319, and on line 409. ***

***We made the suggested edits to address all comments below. New line numbers are noted. ***

3. Line 48: It might be nice to include the range here to support the “high variability” noted here. 

*** (now on line 47). ***

4. Line 54: References cited could be reflected as 12-18 unless there is a typographical error here. 

*** (corrected on line 59) ***

5. Line 108-110: This sentence might benefit by adding a reference. 

*** (A reference supporting the point that recent veterans served multiple and longer tours was added on lines 116.) ***

6. Lines 235-237, Tables 1, 2 and 3: There are 3 different numbers associated with the VA Primary Care-I cohort: 114 (line 235), 131 (table 1) and 104 (tables 2 and 3). It is not clear what these differences are or why they are presented with different denominators.

***Apologies for failing to explain these differences! We have clarified on line 231 that data were collected from 131 veterans and on line 239 that data on stress in relationships related to deployment were collected from the 118 veterans who had been deployed during military service. On lines 254-257, we state that data on Relationship Stress: Spouse/Partner were available for 32 veterans who were still in relationships with the same spouse/partner as at the time of deployment and that data on Relationship Stress: Friends were available for 104 of the 118 veterans who were deployed during military service. *** 

7. Lines 318-319: It appears this sentence is inadvertently cut short. 

***We added the missing word “approach” on line 324. ***

8. Line 332: One iteration of the word “identity” should be “identify.” *** (corrected on line 337) ***

9. Line 380 (in revised manuscript on line 391-393): Is the phrase, “who reported major stressors...” supposed to be, “who reported no major stressors...”? 

***The statement is correct as written. Because new stressors during the follow-up period would be expected to cause symptoms, we EXCLUDED data from those who reported major stressors that were considered potentially traumatic between enrollment and followup. Stated another way, we only included data from those who reported no new potentially traumatic stressors during the follow-up period. *** 

10. Line 381-383: This sentence is important to understanding why Table 4 does not include 232 patients, so it may be better to locate it closer to the tables and the mention of 232 patients (line 395). 

***We added mention of the included data on line 416-417 just before Table 4. ***

11. Table 3: Does not include the Gulf War cohort. Is there a reason for this?

***We added an explanation on lines 277-280. The IA Military sample was used to select items for brief measures of deployment-related variables rather than data from the Gulf War sample because the former was a much larger sample and the data were collected prospectively and over a comparable time frame between risks and PTSD symptoms to be predicted. ***

12. Table 3: For clarity, it might be useful to note that the VA PTSD Resident & Homeless cohort is the combination of two cohorts presented separately in Tables 1 and 2? Is there a reason these are combined here but not elsewhere?

***We added a footnote to Table 3 to explain that the data for VA PTSD Resident and Homeless samples were combined for this analysis because the veterans are very similar and using a larger sample is better to select items for the brief measure of dissociation. ***

13. Table 4: Dissociation is listed as having 4 items here but having 5 items in tables 3 and 5.

***The number in table 4 should be 5. This has been corrected. We are impressed that you noticed this detail! ***

14. Table 5: The caption notes 5 risk factors but only lists 4 in the table. 

***The table name has been corrected to state four risk factors. ***

15. There is a mix of citation styles used in the manuscript that should be unified.

***This has been corrected. ***

16. Abbreviations that are not specified in the text and tables include:

a. PTSD: Line 29 ***Spelled out on line 28***

b. VA: Line 33 ***Spelled out on line 32***

c. VHA: Line 36 ***Changed all instances of VHA to VA for consistency. ***

d. SPTSS: Line 214 ***Spelled out on line 209-210. ***

e. IA (presumably Iraq/Afghanistan): Tables 1, 2, and 3 ***Spelled out in Table footnotes. ***

Thank you again for your work on this manuscript.

Reviewer #2: This is a well-done and important study. I have only a few comments for the authors to consider.

1) The authors make the case that 'moral injury' is a uniquely predictive risk factor for PTSD and Depression and developed 4 novel items to assess moral injury. Because these items were developed de novo for this study, I would recommend that the authors further substantiate their decision to include these items instead of the 'next best' predictive risk factor, e.g., depression or social constraints. I also suggest a bit more description of the construct and/or noting the 4 assessment items developed by the panel since this construct is less well-understood than the other risk factors.

***Thanks for this comment. Your comment made me realize that a better explanation was needed of how the forward regression was conducted. We didn’t choose which variables to enter into the regression, the software did! We have added this explanation on lines 398-401 of the Data Analysis section. Given the results, we agree that the paper needs more explanation of how moral injury is defined and how it was operationalized in the study. On lines 105-111, we have expanded the explanation of moral injury and added a new reference. On lines 358-363, we have included the items used to assess moral injury. On lines 451-454, we have provided a more explicit explanation of why moral injury contributed unique variance to the prediction of later PTSD. *** 

2) The authors note that a limitation of the study is that the population in the cross-validation study is proportionately younger than the population served by the VA in general. It would be helpful for the reader to know to what degree the authors expect that this difference would impact the external validity of their results, i.e., how big of a deal is this? Does this have implications for future studies to confirm the risk factor assessment?

***We gave this comment a great deal of thought and realized that our original point wasn’t sound. While younger veterans who served in combat zones are not representative of all veterans who receive health care at VA, they are representative of veterans who currently present for the first time to VA for health care. For this reason, we decided it was best to omit the original point. *** 

3) On line 319, there appears to be a word missing from the end of the sentence, "...a more patient centered ____". I think this is missing the word "model" or "approach". 

***Yes, the missing word was “approach”! This has been corrected. ***

4) This line (323-324) is a worded a little confusingly: "We examined the predictive or prospective validity of multiple risk factors as this is the form of criterion-related validity that is relevant when predicting future status is the goal".

***We reworded this sentence on line 328-329 to improve its clarity. ***

---

## [Decision Letter · Decision Letter 1]

25 Jul 2022

Development and cross-validation of a veterans mental health risk factor screen

PONE-D-21-33139R1

Dear Dr. Carlson,

We’re pleased to inform you that your manuscript has been judged scientifically suitable for publication and will be formally accepted for publication once it meets all outstanding technical requirements.

Kind regards,

Chong Chen

Academic Editor

PLOS ONE

Additional Editor Comments (optional):

Reviewers' comments:

Reviewer's Responses to Questions

**Comments to the Author**

1. If the authors have adequately addressed your comments raised in a previous round of review and you feel that this manuscript is now acceptable for publication, you may indicate that here to bypass the “Comments to the Author” section, enter your conflict of interest statement in the “Confidential to Editor” section, and submit your "Accept" recommendation.

Reviewer #1: All comments have been addressed

Reviewer #2: All comments have been addressed

2. Is the manuscript technically sound, and do the data support the conclusions?

Reviewer #1: Yes

Reviewer #2: Yes

3. Has the statistical analysis been performed appropriately and rigorously? 

Reviewer #1: Yes

Reviewer #2: Yes

4. Have the authors made all data underlying the findings in their manuscript fully available?

Reviewer #1: Yes

Reviewer #2: Yes

5. Is the manuscript presented in an intelligible fashion and written in standard English?

Reviewer #1: Yes

Reviewer #2: Yes

6. Review Comments to the Author

Reviewer #1: The authors are to be commended for addressing the items laid out by the reviewers. I found the paper much easier to digest and stronger as a result of the effort made. Overall, I recommend it be accepted for publication.

Reviewer #2: This is a fantastic piece of scholarship. Thank you for your contribution to the literature and the opportunity to review this work.

7. PLOS authors have the option to publish the peer review history of their article (what does this mean?). If published, this will include your full peer review and any attached files.

Reviewer #1: No

Reviewer #2: **Yes: **Lauren E. Allen, DrPH

---

## [Editor Report · Acceptance letter]

6 Oct 2022

PONE-D-21-33139R1 

Development and cross-validation of a veterans mental health risk factor screen 

Dear Dr. Carlson:

I'm pleased to inform you that your manuscript has been deemed suitable for publication in PLOS ONE. Congratulations! Your manuscript is now with our production department. 

Kind regards, 

on behalf of

Dr. Chong Chen 

Academic Editor

PLOS ONE